# How Would the Potential Collapse of the Cumbre Vieja Volcano in La Palma Canary Islands Impact the Guadeloupe Islands? Insights into the Consequences of Climate Change

Gael E. Arnaud [1,*], Yann Krien [2], Stéphane Abadie [3], Narcisse Zahibo [4] and Bernard Dudon [4]

1   MetOcean Solutions Ltd., 5 Wainui Road, Raglan 3225, New Zealand
2   Littoral ENvironnement et Sociétés—UMR 7266, 17000 La Rochelle, France; yann.krien@univ-lr.fr
3   Université de Pau et des Pays de l'Adour, E2S UPPA, SIAME, 64600 Anglet, France; stephane.abadie@univ-pau.fr
4   Laboratoire de Recherche en Géosciences et Energies, Université des Antilles, Pointe-à-Pitre 97157, Guadeloupe; narcisse.zahibo@univ-antilles.fr (N.Z.); bdudon@gmail.com (B.D.)
*   Correspondence: gael.arnaud@metocean.co.nz

**Abstract:** Tsunamis are among the deadliest threats to coastal areas as reminded by the recent tragic events in the Indian Ocean in 2004 and in Japan in 2011. A large number of tropical islands are indeed exposed due to their proximity to potential tsunami sources in tectonic subduction zones. For these territories, assessing tsunamis' impact is of major concern for early warning systems and management plans. The effectiveness of inundation predictions relies, among other things, on processes engaged at the scale of the local bathymetry and topography. As part of the project C3AF that aimed to study the consequences of climate change on the French West Indies, we used the numerical model SCHISM to simulate the propagation of several potential tsunamis as well as their impacts on the Guadeloupe islands (French West Indies). Working from the findings of the most recent studies, we used the simulations of four scenarios of collapse of the Cumbre Vieja volcano in La Palma, Canary islands. We then used FUNWAVE-TVD to simulate trans-Atlantic wave propagation until they reached the Guadeloupe archipelago where we used SCHISM to assess their final impact. Inundation is quantified for the whole archipelago and detailed for the most exposed areas. Finally, in a climate change perspective, inundation is compared for different sea levels and degrees of vegetation cover deterioration using modified friction coefficients. We then discuss the results showing that climate change-related factors would amplify the impact more in the case of smaller inundation along with model limitations and assumptions.

**Keywords:** tsunami; SCHISM; Guadeloupe; Cumbre Vieja; climate change

## 1. Introduction

Tsunamis are among the deadliest threats to coastal areas. The recent events in the Indian Ocean in 2004 or in Japan in 2011 show that the number of fatalities can dramatically increase depending on whether coastal areas and their population are prepared or not. The Tōhoku tsunami, in 2011 killed almost 20,000 people [1] while more than 220,000 victims have been reported after the Sumatra-Andaman tsunami of 2004 in the Indian Ocean [2]. This difference in human losses evidences the importance of preparedness to such events and highlights the need for accurate hazard assessment.

In the Caribbean, islands populations are particularly exposed to tsunamis hazards and significant efforts have been made in recent decades to identify potential sources since the presence of subduction zones near the Lesser Antilles creates a high potential of tsunami hazard [3–6]. As part of the C3AF project (Consequences of Climate Change in the French Antilles) several scenarios of potential tsunamis have been simulated to develop tsunami inundation maps for the Guadeloupe archipelago. These maps have been created for the

attention of local authorities and councils, to assist with urban planning and evacuation plan design. Among the wide range of potential sources, distant sources have also been considered since teletsunamis may also have important consequences along the Caribbean coasts [7,8]. Among such distant sources, the collapse of the Cumbre Vieja volcano, Canary islands, is recognized as a serious potential threat for the Atlantic Basin and was the object of several studies ([9–12], among others). Although these studies present some simulations of tsunami propagation, across the Atlantic Basin, they mainly focused on propagation toward the North American, European and African coasts as well as toward some nearby islands [13–15] while only scarce information is given for the Lesser Antilles.

### 1.1. The Collapse of the Cumbre Vieja Volcano and Tsunamis

Despite its name, the Cumbre Vieja is a rather young volcano and the fastest growing in the Canary islands. It is thus considered as posing the largest threat of collapse in these islands [16]. Ward and Day [10] made a pioneering work on a potential tsunami generated by the collapse of the Cumbre Vieja volcano's Western flank on La Palma island, Canary. With an assumed slide volume of 500 km$^3$, they found that the waves generated by the collapse would potentially hit the east coast of North America with a height in the range of 10–25 m. This extreme scenario has been controversial and was later contested, first because of the geological aspect of the slide [17,18] and second because of the modelling of the propagation [11,12,19]. Gisler et al. [19] used a 3D Navier–Stokes equation set to model the slide and consecutive wave. Far field was then estimated using an extrapolation of near field decay, and the authors concluded like Mader [11] that wave height would not represent a serious threat for the east coast of North America or South America. Løvholt et al. [12] started from the near field solution presented by Gisler et al. [19] and simulated transoceanic propagation using a Boussinesq method to factor in dispersive effects. They found smaller waves than Ward and Day [10] but that were still potentially dangerous. Their conclusion also emphasized the need for dispersive equations to correctly represent both decay and wave shape during transoceanic propagation. Later, Abadie et al. [9] proposed a similar method with a 3D, multiphase, Navier–Stokes equation set to model the landslide. Because of the uncertain likelihood of the event, they proposed four different slide volumes, ranging from 20 to 450 km$^3$. Starting from these collapse simulations, Tehranirad et al. [15] simulated the propagation across the Atlantic Basin towards the East coast of the U.S. to assess the inundation in this location. Recently, Abadie et al. [20] reviewed their collapse modelling, using a greater viscosity to better approximate a granular behaviour. Using these new results, propagation has been simulated towards Europe and Guadeloupe and impact assessments have been conducted for several regions of interest. Results have shown, that even according to the most minimal scenario, the Guadeloupe archipelago can be affected and the impact would be potentially catastrophic if the collapsed volume exceeded 40 km$^3$.

Based on these scenarios, this study details the potential inundation for the whole archipelago, identifying the most threatened areas. Additionally, using assumption on sea level rise to assess the evolution of the threat, we examine and analyse several aspects of climate change.

## 2. Method

### 2.1. The Guadeloupe Archipelago

The Guadeloupe archipelago is located 61° W and 16° N in the Lesser Antilles, 4600 km west-south-west of the Cumbre Vieja volcano. It is made of four main groups of islands (Figure 1) with a total surface of 1628 km$^2$. The two main islands, Basse-Terre (volcanic, mountainous with steep slopes) and Grande-Terre (sedimentary flat uplifted) are connected by a low-lying pseudoisthmus cut across by a salt water river. The capital Basse-Terre (town) is located on the south western side of the Basse-Terre island. The economic activity area (port, power plant, airport) is located in the central part of the isthmus near the town of Point-à-Pitre. The shelf is globally narrow (less than 2 km wide) but with a larger part

off St François where it spreads to a width of about 10 km to the east-south-east to join the island of La Désirade. A large semicircular shelf of about 20 km in diameter also surrounds The Saintes islands, located slightly off-centre to the west. Both sides of the pseudoisthmus between the two main islands are flanked by wide shallow lagoons (1–5 m deep). The northern part is more developed and sheltered by a coral barrier, which sits 5–10 km away from the coast. According to Comte [21], 107 km$^2$ of the whole archipelago are potentially subjected to coastal floods, most of which are located on Grande-Terre. Tide is is mixed semidiurnal with strong seasonal variations (≈15 cm) with a small tidal range (a few dozen centimetres). Seiches of several centimetres are also observed on tidal records such as, the Pointe-à-Pitre tide gauge [22]. Although their origin is not well understood, Woodworth [23] suggests that oscillation periods may indicate the frequencies at which sea level will oscillate if a tsunami occurs.

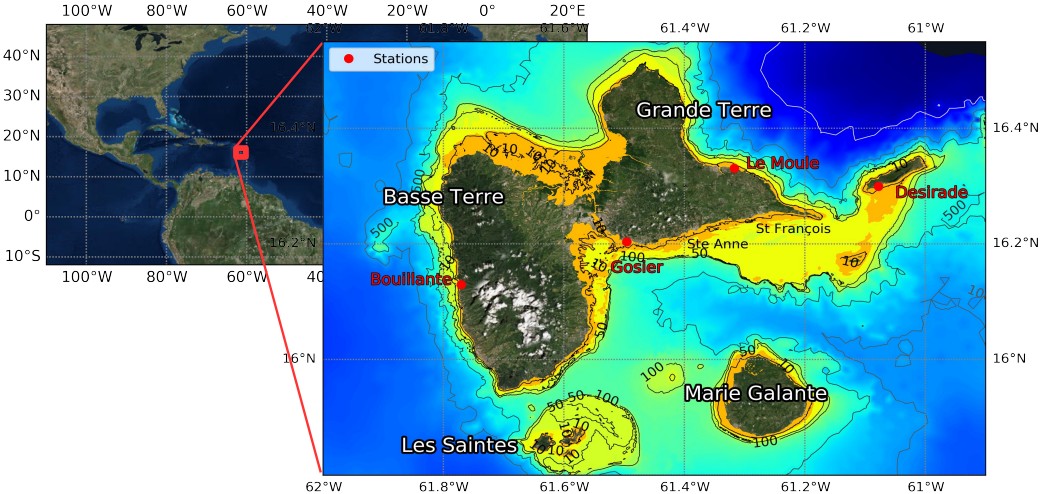

**Figure 1.** Guadeloupe archipelago location and bathymetry. Virtual gauges are indicated by red dots.

### 2.2. Model Implementation

Tsunamis can generally be be broken down into three stages: generation, propagation and final inundation. The generation phase is a crucial point of modelling depending on the type of source considered. For the purposes of this study, we have not modelled the source but relied for this stage on a new computation by Abadie et al. [20], which is an update of the simulations presented in Abadie et al. [9] with a higher viscosity so as to better reproduce a granular slide. They used viscosity values obtained through a calibration according to experimental results such as Viroulet et al. [24]. The new simulation of Abadie et al. [20] focuses on the first 20 min of the event. The collapse of the flank of the volcano and subsequent landslide (generation) are modelled over 5 min, using a full Navier–Stoke multiphase approach and considering the sliding material and the water as distinct fluids. This is followed by a 15 min simulation of wave propagation in the vicinity of the collapse of the volcano's flank, using a Boussinesq-type model over a 500 m grid resolution. We simulated different scenarios based on four different landslide volumes: 20 km$^3$ and 40 km$^3$ which represent a partial failure of the Volcano's flank; 80 km$^3$ which is the reference case noted as the extreme credible worst-case scenario; and an extreme 450 km$^3$ scenario to be used as a comparison with previous studies [10,12].

Our study works from these first stages of simulation to propagate the waves towards Guadeloupe and assess their impact on the archipelago. This transatlantic propagation has already been studied in depth [12,19,25]: the conclusions emphasised that dispersive effects are still significant for a large part of the transatlantic propagation and hence according to Løvholt et al. [12], Linear Shallow Water equations and Nonlinear Shallow Water equations should not be used here. To take these effects into account, we modelled the trans-Atlantic

propagation using a Boussinesq-type model (FUNWAVE-TVD) initiated from the 15 min simulation of FUNWAVE-TVD in the vicinity of the volcano's collapse.

### 2.3. Trans-Oceanic Model

Trans-Atlantic propagation is simulated using the FUNWAVE-TVD code [26], the most recent implementation of the Boussinesq model FUNWAVE [27], initially developed and extensively validated for nearshore wave processes but also used to perform tsunami case studies [15,28,29]. The code, solves the Boussinesq equations of Chen [30] with either fully nonlinear equations in a Cartesian framework [26] or a weakly nonlinear spherical coordinate formulation with Coriolis effects [31]. In this case, for the convenience of a large oceanic grid covering the Atlantic ocean, the code is used with spherical coordinates configuration on a regular grid with a one-minute arc resolution ($\approx$1.8 km) over a grid of 3301 $\times$ 1681 cells. The computational domain covers a large part of the Northern Atlantic Ocean between 11° N and 41° N as presented on Figure 2. At t = 18,900 s (5 h 30 after the volcano collapse) when the front wave has crossed the Atlantic Ocean and is about 180 km east of La Désirade, the FUNWAVE state is interpolated over the last model grid and inputted as a hotstart with a 2-D water level and horizontal velocity field considering the cases of 20, 40, 80 and 450 km$^3$ slide volumes.

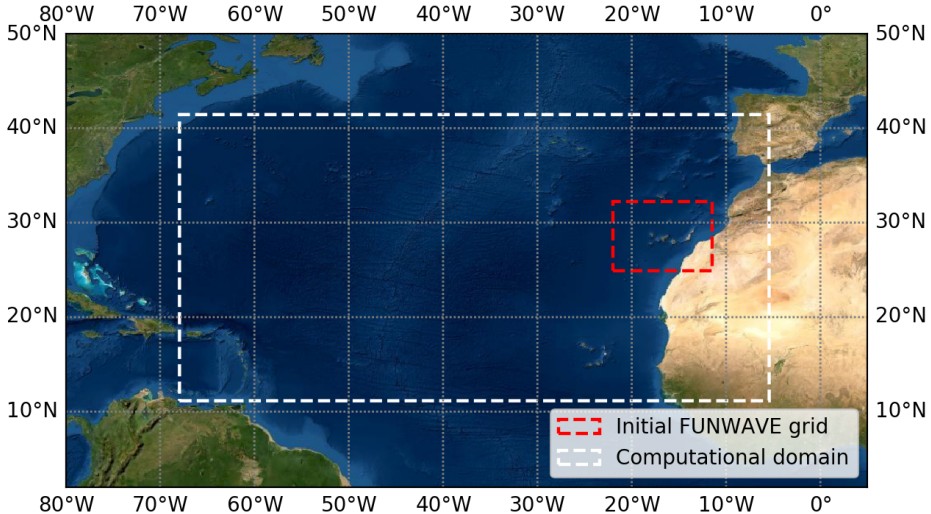

**Figure 2.** Computational domains. The red area represents the first 20 min of FUNWAVE simulation over a 500 m grid resolution. The white area delimits the FUNWAVE domain for the trans-Atlantic propagation with 1 arc degree resolution, as well as the SCHISM grid domain where the hotstart takes place at 5:40.

### 2.4. Inundation Model

The last stage of the simulation covers the propagation in the nearshore and inundation. As in shallow water nonlinear processes take precedence over dispersive effects, this stage relies on the use of the circulation model SCHISM (Semi-implicit Cross-scale Hydroscience Integrated System Model) [32], a derivative product of SELFE [33].

Although the code is able to solve the 3D Reynolds Averaged Navier-Stokes equations in hydrostatic or nonhydrostatic mode, in this study, we only use one sigma layer which leads to depth-averaged NLSW equations in hydrostatic and barotropic mode. These equations are solved over an unstructured mesh which allows for an accurate representation of coast line features. Along the studied coastline, we achieve a resolution of 8 m, including in the aerial domain where specific features may obstruct the water flow inland. The flooding process is modelised using a specific inundation algorithm, which is detailed and bench-marked in Zhang and Baptista [34] and avoids large elevation gradients that would

lead to unrealistically large velocities. The simulations are initiated with a hotstart from FUNWAVE state at t = 18,900 s.

### 2.4.1. Grid

The inundation process is a crucial step: particular attention has been brought to the definition of topographic features, to make sure that all obstacles are represented in the grid. Topography and bathymetry in shallow water up to 40 m were specified using high resolution LiDAR bare-earth Digital Elevation Model (DEM) available from the Litto3D program (SHOM & IGN). For deeper waters, the General Bathymetric Chart of the Oceans (GEBCO) was used with 30 arcsec resolution. The unstructured grid incorporates feature lines into the meshing pattern to ensure that these features are considered (coast-line at 0 m, crest of sea walls, mounts etc.) Moreover, grid resolution is also finer where topography gradients are steep in the aerial domain, considering the banks that may drive water flows.

### 2.4.2. Bottom Friction

Bottom friction is computed in SCHISM assuming a quadratic function with a Manning-Strickler coefficient. To better represent disparities, this coefficient may vary according to bottom roughness. Since the DEM is a bare-earth model, an adapted friction coefficient is applied according to the cover type in order to account for buildings, vegetation and other obstacles that cannot be individually represented in the grid. The distribution of these friction values (Figure 3) is based on the Corine Land Cover dataset [35] available for the Guadeloupe in which land use is defined under the code CLC level 4, specific to overseas French territories as shown in Table 1. The Manning value is then set according the prior studies by Chow [36] and using guidelines of Arcement and Schneider [37]. The distribution of submarine friction coefficients is based on the work of Chauvaud et al. [38] which defines 6 classes within the marine biocenosis in the shallow waters surrounding the Guadeloupe archipelago. Consequently, Manning values were set according to the type of biocenosis (Table 2). The Manning values thus defined range from 0.022 s/m$^{1/3}$ to 0.2 s/m$^{1/3}$ (Figure 3).

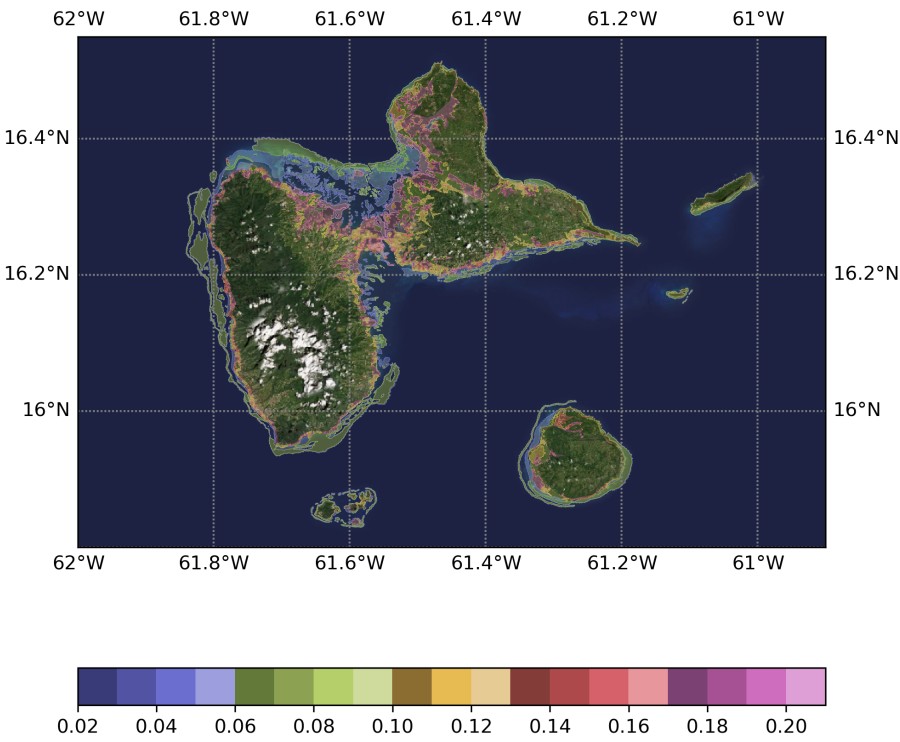

**Figure 3.** Manning-Strickler coefficient values for the SCHISM grid over the Guadeloupe archipelago for subaerial and aerial domain.

**Table 1.** Table of Manning coefficients for the aerial coverage according to the Corinne Land Cover classification.

| CLCL 4 | Manning | Description |
|--------|---------|-------------|
| 1110 | 0.15 | Continuous urban fabric |
| 1120 | 0.12 | Discontinuous urban fabric |
| 1210 | 0.15 | Industrial or commercial units and public facilities |
| 1220 | 0.03 | Road and rail networks and associated land |
| 1230 | 0.05 | Port areas |
| 1240 | 0.02 | Airports |
| 1310 | 0.12 | Mineral extraction sites |
| 1320 | 0.12 | Dump sites |
| 1330 | 0.12 | Construction sites |
| 1410 | 0.04 | Green urban areas |
| 1420 | 0.04 | Sport and leisure facilities |
| 2111 | 0.06 | Nonirrigated arable land |
| 2112 | 0.18 | Sugar cane |
| 2120 | 0.04 | Permanently irrigated land |
| 2130 | 0.04 | Rice fields |
| 2210 | 0.04 | Vineyards |
| 2221 | 0.15 | Fruit trees and berry plantations |
| 2222 | 0.18 | Banana plantations |
| 2223 | 0.18 | Palm groves |
| 2224 | 0.18 | Coffee trees |
| 2230 | 0.09 | Olive groves |
| 2310 | 0.06 | Pastures, meadows and other permanent grasslands under agricultural use |
| 2410 | 0.06 | Annual crops associated with permanent crops |
| 2420 | 0.06 | Complex cultivation patterns |
| 2430 | 0.06 | Land principally occupied by agriculture, with significant areas of natural vegetation |
| 2440 | 0.12 | Agro-forestry areas |
| 3111 | 0.18 | Broad-leaved forest |
| 3112 | 0.2 | Mangrove forest |
| 3120 | 0.09 | Coniferous forest |
| 3130 | 0.18 | Mixed forest |
| 3210 | 0.04 | Natural grasslands |
| 3220 | 0.18 | Moors and heathland |
| 3230 | 0.1 | Sclerophyllous vegetation |
| 3240 | 0.15 | Transitional woodland-shrub |
| 3310 | 0.03 | Beaches, dunes, sands |
| 3320 | 0.03 | Bare rocks |
| 3330 | 0.04 | Sparsely vegetated areas |
| 3340 | 0.04 | Burnt areas |
| 3350 | 0.025 | Glaciers and perpetual snow |
| 4110 | 0.12 | Inland marshes |
| 4120 | 0.025 | Peat bogs |
| 4210 | 0.04 | Coastal salt marshes |
| 4220 | 0.03 | Salines |
| 4230 | 0.03 | Intertidal flats |
| 5111 | 0.025 | Water courses |
| 5112 | 0.025 | Temporary water course |
| 5120 | 0.025 | Water bodies |
| 5210 | 0.025 | Coastal lagoons |
| 5220 | 0.025 | Estuaries |
| 5230 | 0.025 | Sea and ocean |

**Table 2.** Table of Manning coefficients for coastal shallow water according to classification by Chauvaud et al. [38].

| Manning | Description |
| --- | --- |
| 0.04 | Meadows |
| 0.09 | coral and other bentic habitats |
| 0.06 | coral and other bentic habitats—Meadows |
| 0.07 | coral and other bentic habitats—Seaweed |
| 0.04 | Meadows—Seaweed |
| 0.03 | Seaweed |

*2.5. Climate Change Scenarios*

The impact of climate change is assessed by investigating two parameters in our simulations to mimic the forcing of climate changes that may act upon tsunami processes. The first parameter is Sea Level Rise (SLR) since the region is expected to experience a rise of about 70 cm by 2100 Oppenheimer et al. [39]. According to these projections, the scenario adopted here would be a rise of 80 cm of the mean water level. The four landslide scenarios are thus simulated with the current mean sea level and then run again with a 80 cm rise of initial water levels to assess and quantify the changes in impact The second parameter is the speculated deterioration of the mangrove which can go along with the deterioration of the coral reef and other natural protections. We know that vegetation can play a crucial role in the mitigation of flooding processes. Moreover, human activities have an undoubtable impact on biodiversity with unfortunate consequences that can lead to the elimination of this natural barrier. Liu et al. [40] investigated the friction in a mangrove area in Taiwan and estimate a Manning friction coefficient of between 0.088 and 0.28 $s/m^{1/3}$. Therefore, to assess the degradation, the friction coefficient used within the CLC-4 polygons representing the mangrove is gradually reduced from a maximum value of 0.2 (Manning configuration 2) for a healthy and dense mangrove, to a medium value of 0.14 (Manning configuration 3) for a slightly deteriorated mangrove and to 0.8 (Manning configuration 4) that would represent a deteriorated mangrove.

**3. Results**

*3.1. Time of Arrivals and Wave Heights*

Arrival times and wave heights are shown in Figure 4 for the four stations located on Figure 1. For the sake of clarity, water levels for the 450 $km^3$ scenario are not displayed on charts since heights for this extreme scenario can be up to seven times higher than those for the 80 $km^3$ scenario (i.e., 12 m at La Désirade, 20 m at Le Moule, 15 m at Le Gosier and 10 m at Bouillante.) Travelling time is a just under 6 h and a minimal time delay is observed between stations. The eastern parts, Le Moule and La Désirade are immediately exposed to the incoming wave front and thus are the first hit, respectively at 5 h 48 min and 5 h 57 min after the event. They are followed by the northern and southern coasts as the wave wraps around Grande-Terre. Le Gosier is hit after 6 h 10 min and the time difference between the easterly point and the leeward side is about 15 min. The first waves on the leeward side are not the tallest and the maximum height comes in 40 min later. It is also noticed that the wave period the leeward side is twice shorter than the period of waves coming in straight windward.

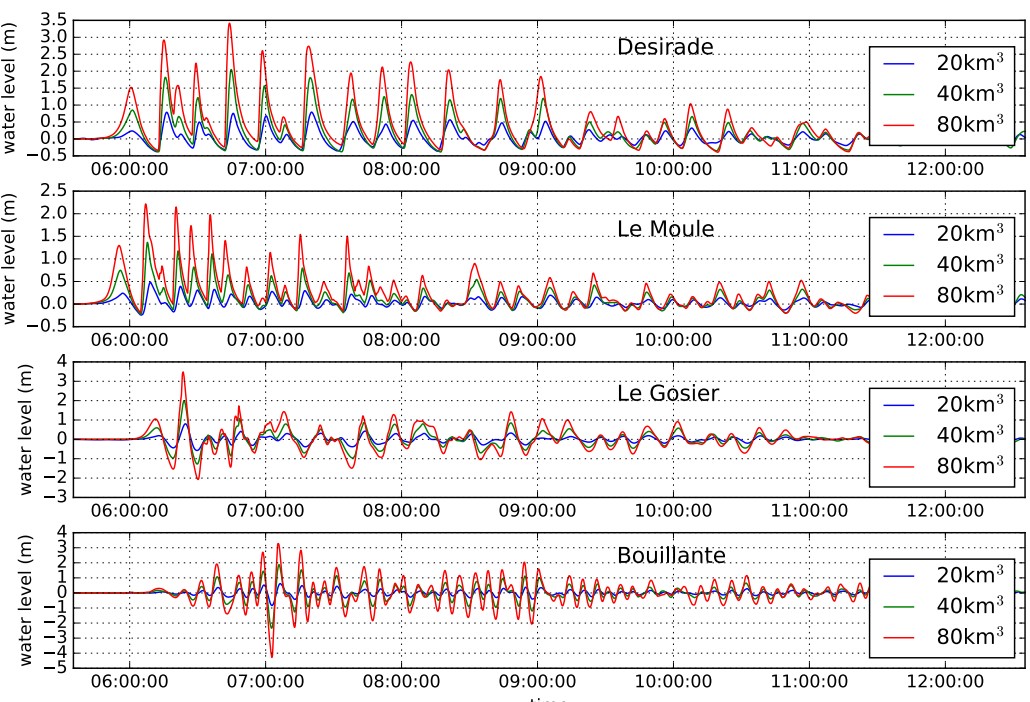

**Figure 4.** Arrival time after the collapse (including the 20 min initial simulation) at the stations located on Figure 1 and water elevation for the 3 cases 20, 40 and 80 km$^3$.

### 3.2. Wave Heights Distribution

Water level is expressed as maximum elevation for all landslide scenarios and through the whole simulation process on Figure 5. For each scenario, the surface of flooded land is mentioned in km$^2$—this information is used below in Section 3.4.1. The distribution shows the same patterns for all scenarios and differs only in wave amplitude. The waves are refracted and driven by the bathymetry. The most exposed areas are Le Moule, La Désirade, Saint-François, Marie-Galante, Les Saintes and Sainte-Anne, mainly located on coasts facing the open ocean but not only: some sheltered sites such as Terre-de-Haut, Les Saintes are also particularly exposed. Offshore, wave heights for the most exposed area are about 2 m for the 20 km$^3$ scenario, 3 m for 40 km$^3$, 5 m for 80 km$^3$ and above 20 m for 450 km$^3$. The large lagoon on the northern side of the central pseudoisthmus is sheltered by a coral reef barrier which is evidenced by the water level difference between the inside and the outside of the lagoon. The other lagoon, smaller, on the southern side of the pseudoisthmus seems to be less sheltered and displays higher water levels.

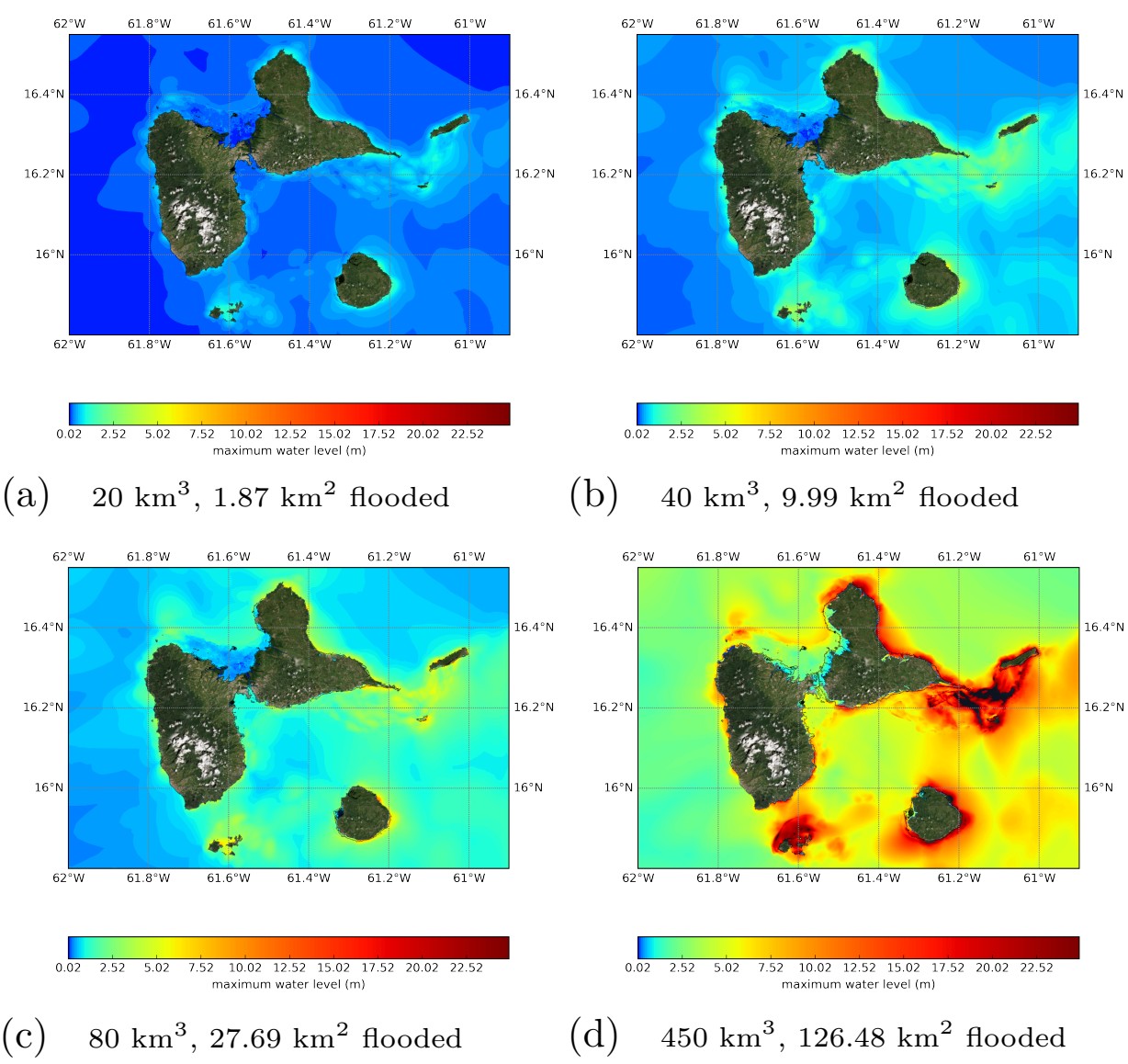

(a)   20 km$^3$, 1.87 km$^2$ flooded

(b)   40 km$^3$, 9.99 km$^2$ flooded

(c)   80 km$^3$, 27.69 km$^2$ flooded

(d)   450 km$^3$, 126.48 km$^2$ flooded

**Figure 5.** Maximum water level for each scenario for the present situation.

### 3.3. Inundation

Considering a general overview, the 20 km$^3$ scenario would only generate minor overtopping in exposed harbours such as La Désirade or Deshaies while the town of Terre-de-Haut is already threatened with this scenario and would be flooded in all other cases. With the 40 km$^3$ scenario, the large majority of coastal towns in Grande-Terre are subjected to floods except for Port Louis and Anse Bertrand. Marie-Galante also shows some quite important floods in Saint Louis, Grand-Bourg and Capesterre. With the 80 km$^3$ scenario, the impact is largely severe; damage would be heavy in some places such as Saint-François (Figures 6 and 7) and globally in all coastal towns in Grande-Terre and Les Saintes, all towns in Marie-Galante, as well as Deshaies and Bouillante on the Basse-Terre island. We note that Pointe-à-Pitre (economic hub) and the town of Basse-Terre (capital town) are relatively spared for scenarios under 80 km$^3$ and inundation in the industrial estate of Baie-Mahault (Pointe-à-Pitre), is quite limited. This would not be the case for the extreme 450 km$^3$ scenario, for which more than 130 km$^2$ of land would be flooded and swept away.

The detailed maps of inundation are presented in the C3AF project's final report and form a large catalogue of maps that cannot all be displayed here. As an example, the town of Saint Francois (eastern Grande-Terre) is presented on Figures 6 and 7 with maximum water depth for each present scenario.

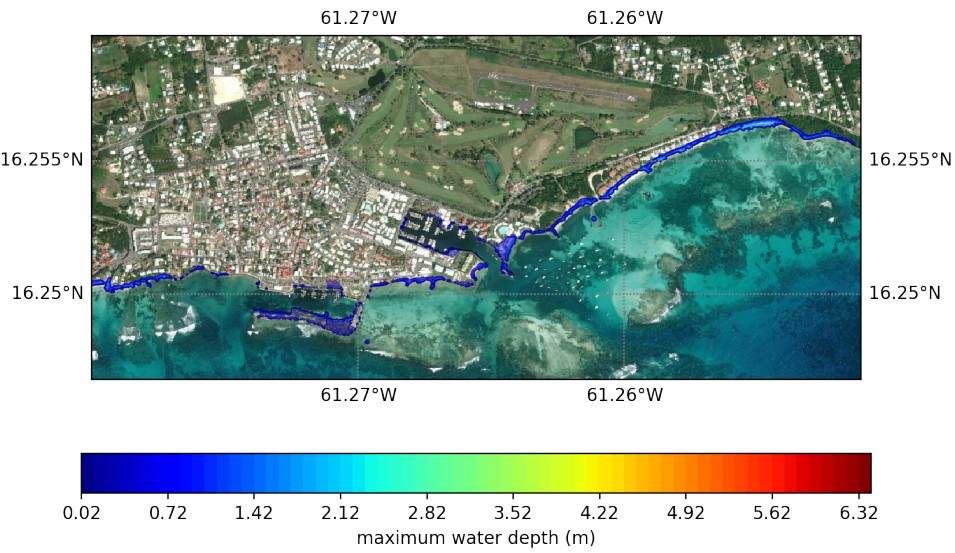

(a)    20 km$^3$

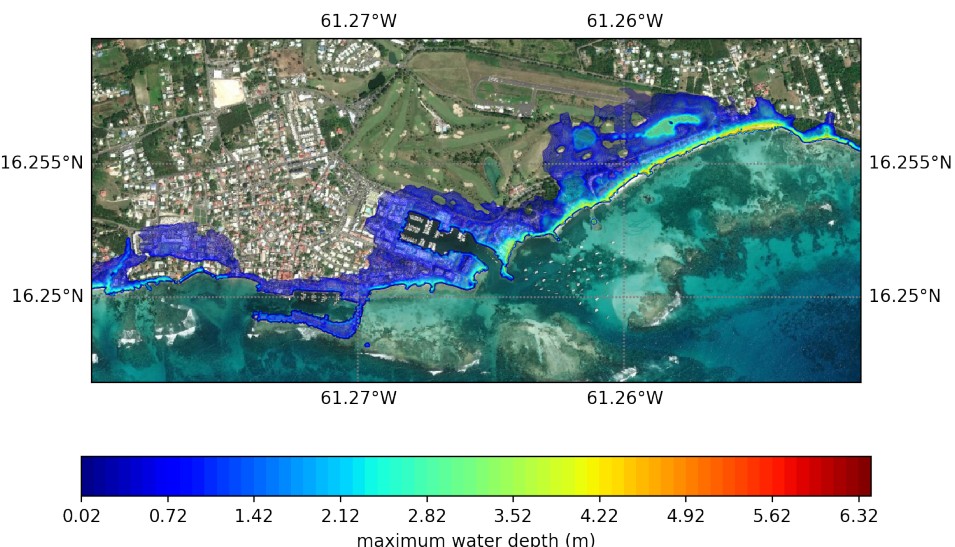

(b)    40 km$^3$

**Figure 6.** Maximum water depth in the area of Saint-François, Grande-Terre for the 20 and 40 km$^3$ scenarios.

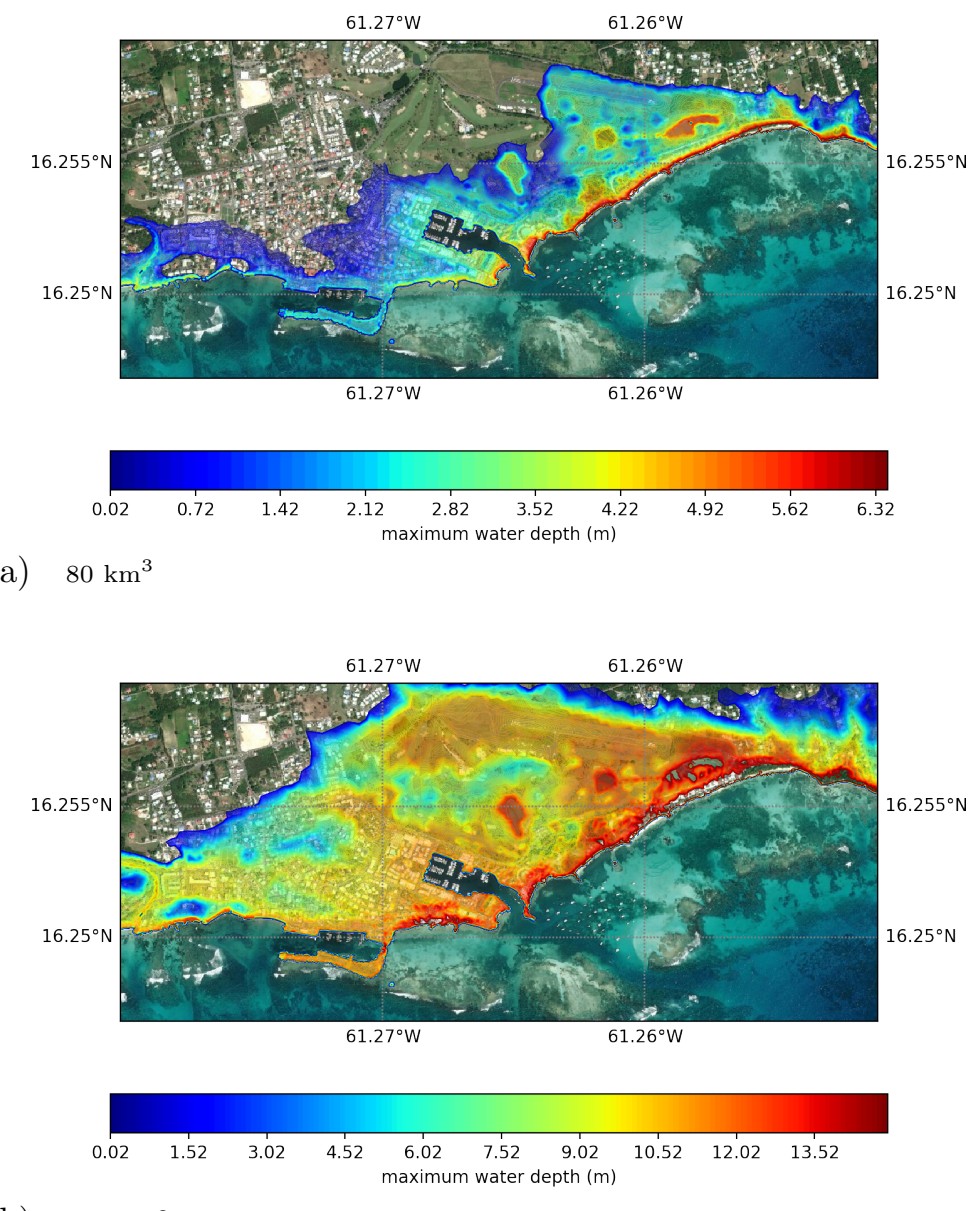

**Figure 7.** Maximum water depth in the area of Saint-François, Grande-Terre for the 80 and 450 km³ scenarios.

As in most places, the 20 km² scenario is not a major threat for the town of Saint-François, although appropriate measures should be taken since the coast line may experience some overtopping. The 40 km³ scenario already displays an inundation of about 1 m in some places around the marina and urban areas close to the coast. An 80 km³ collapse would flood all the houses along the coast and severely impact the town with water above human height inducing important damage and potential casualties if no evacuation measures are taken. The extreme 450 km³ scenario would lead to a flooding of the entire town with a water depth above 10 m in some places.

*3.4. Climate Change-Related Tsunami Impact*

The consequences of climate change are assessed from the variation in impact between currents and future configurations (cf. Section 2.5). The current configuration is considered with no SLR and the same Manning distribution as presented in Table 1. For all scenarios, SLR induces a significant increase of the flooded surface. As an example, Figures 8 and 9

show the maximum water depth reached with a 40 km³ scenario in Grand Baie, Grande-Terre for the current situation (Figure 8) and with an 80 cm water level elevation (Figure 9). The extension of the flooded surface is visible here. The residential area in the north of the bay is indeed severely impacted according the future scenarios while only the first row of houses is impacted in the current situation.

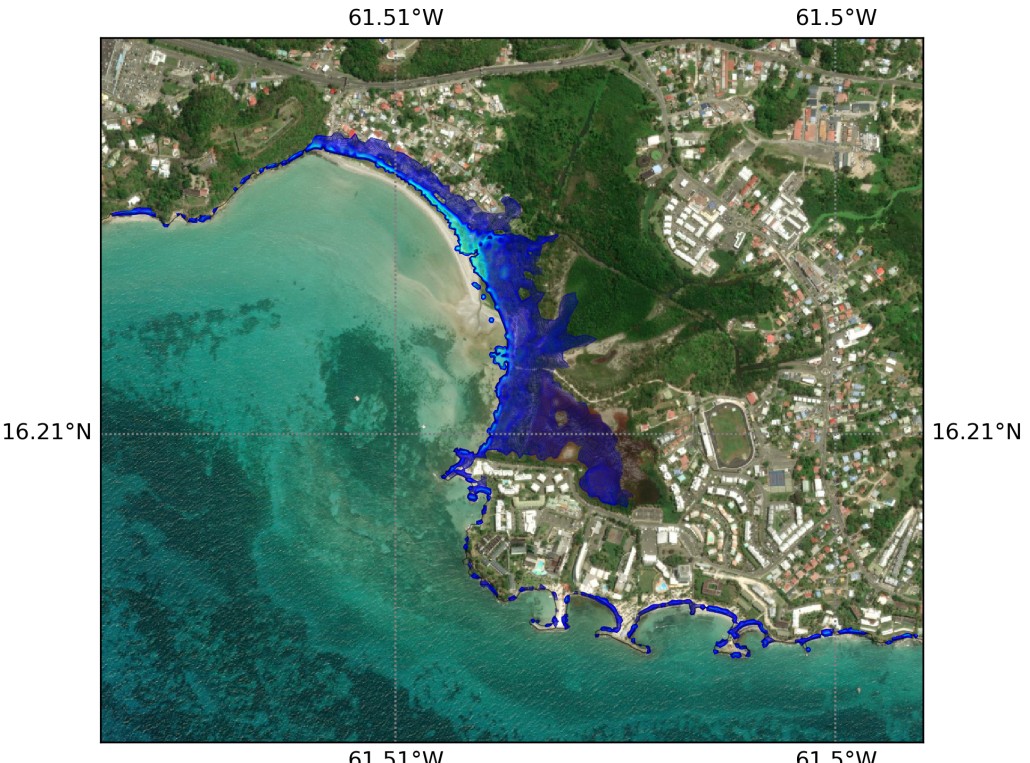

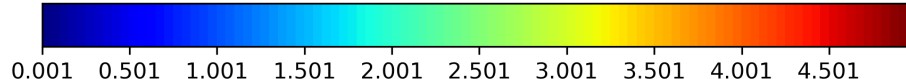

**Figure 8.** Maximum water depth in Grand Baie, Grande-Terre for the 40 km³ scenario in the present configurations.

The town of Pointe-à-Pitre and the port facilities are presented here, as these are the sites where the island's main economic assets are located (power plant, airport, hospital etc.). This location is quite sheltered in the natural harbour and thus provides a good example of impact variations according to future scenarios. In the current configuration, the 20 and 40 km³ scenarios only have a minor impact in this area and even the impact for the 80 km³ is not as severe as in other exposed places. Figure 10 shows the maximum water depth for the 80 km³ scenario where the impact is mostly limited quite limited to the shore and the mangrove along the coastline. However, when considering the sea level rise assumption (Figure 11), the impact of the same scenario is significantly increased: inundation is no longer limited to the shore and progresses further inland to reach the town of Pointe-à-Pitre in the Lauricisque area. The port facilities are also impacted as well as the industrial estate of Baie-Mahault (west bank of the harbour).

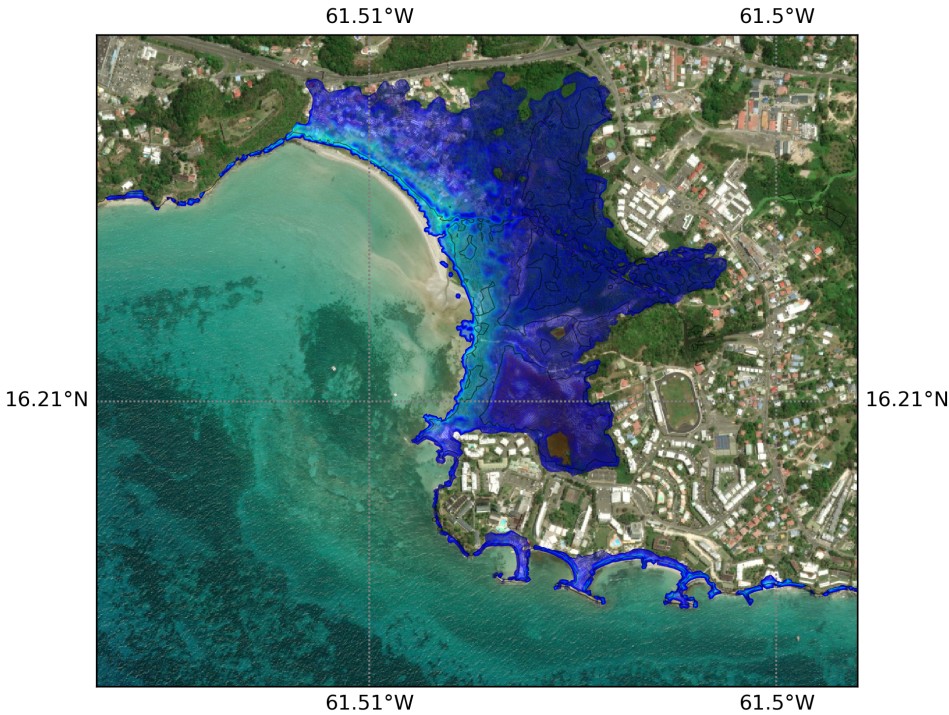

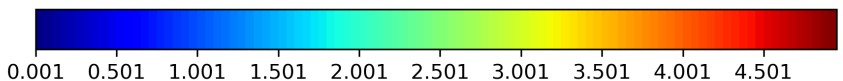

**Figure 9.** Maximum water depth in Grand Baie, Grande-Terre for the 40 km$^3$ scenario in the future configurations.

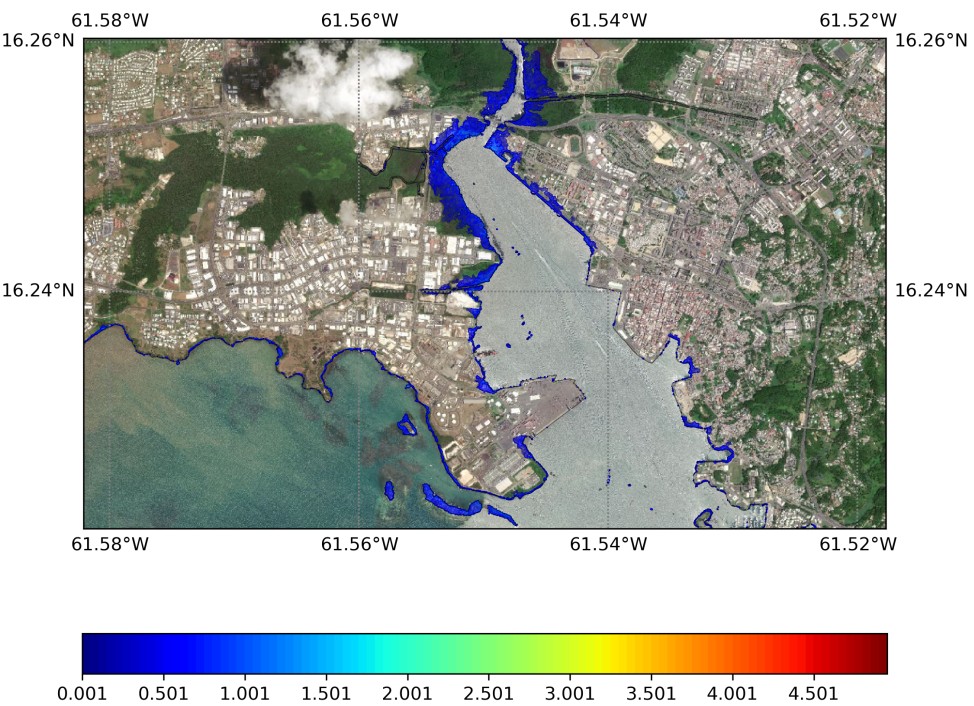

**Figure 10.** Maximum water depth for the Pointe-à-Pitre and Baie-Mahault region, Grande-Terre for the 80 km$^3$ scenario in the present configurations.

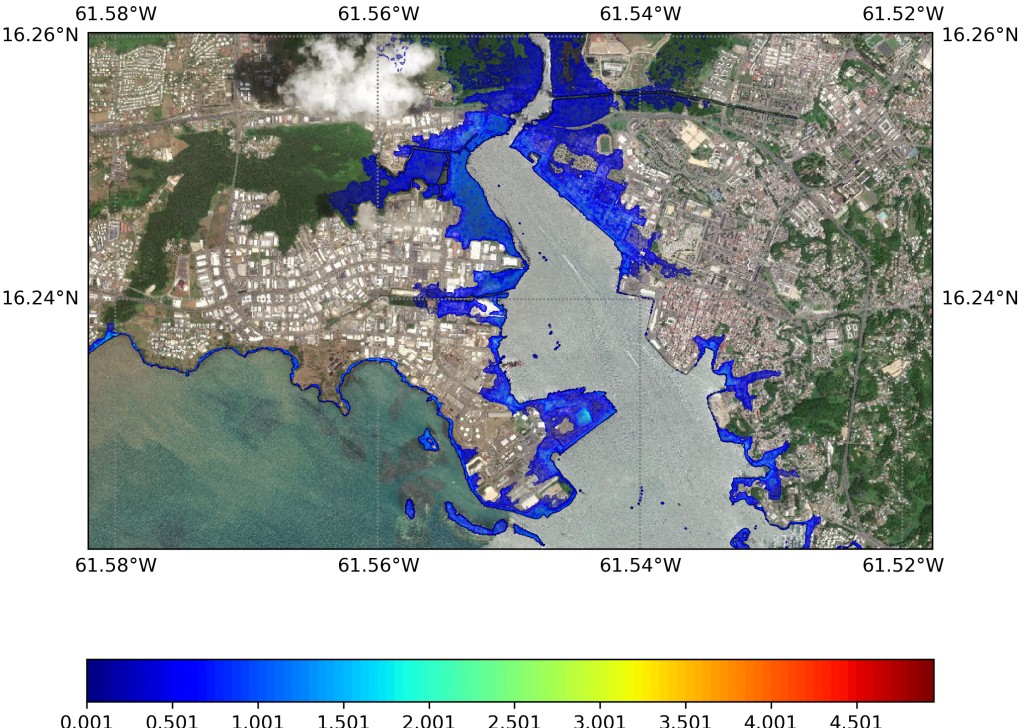

**Figure 11.** Maximum water depth for the Pointe-à-Pitre and Baie-Mahault region, Grande-Terre for the 80 km$^3$ scenario with Sea Level Rise.

3.4.1. Flooded Surfaces Analysis

To quantify climate change impact over the whole archipelago, the flooded surfaces of land are computed to allow for a comparison between simulations. These surfaces are proportional to cells size for which a threshold of 20 cm is set to identify the ones that should be accounted for flooded. The accuracy of the flooded surface is thus directly linked to grid resolution since a cell can only be considered as flooded or not.

The outlines are quite intuitive, as the surface increases with the volume of collapse: the bigger the scenario, the larger the flooded surface. The increase is more or less linear up to the 80 km$^2$ scenario, after which the increase is much greater for the last scenario (450 km$^3$). Inundated surfaces for the 20, 40, 80 and 450 km$^3$ scenarios are respectively 1.87, 9.99, 27.69 and 126.48 km$^2$, as shown on Figure 5.

Variations of inundated areas are reported on Figure 12. Each point shows the difference between flooded surface before and after the climate change assumption. This value is expressed in percentage of increase in flooded surface before sea level rise for the same scenario.

Across all simulations, due to SLR, the flooded surface would increase, from 5% for the largest collapse scenarios to about 25% for the lowest. The differences between Manning configurations, which represent the degradation of natural protection, increase slightly with the volume of the collapse. The 20 km$^3$ scenario is not affected, while the most affected scenarios are those for 40 and 80 km$^3$.

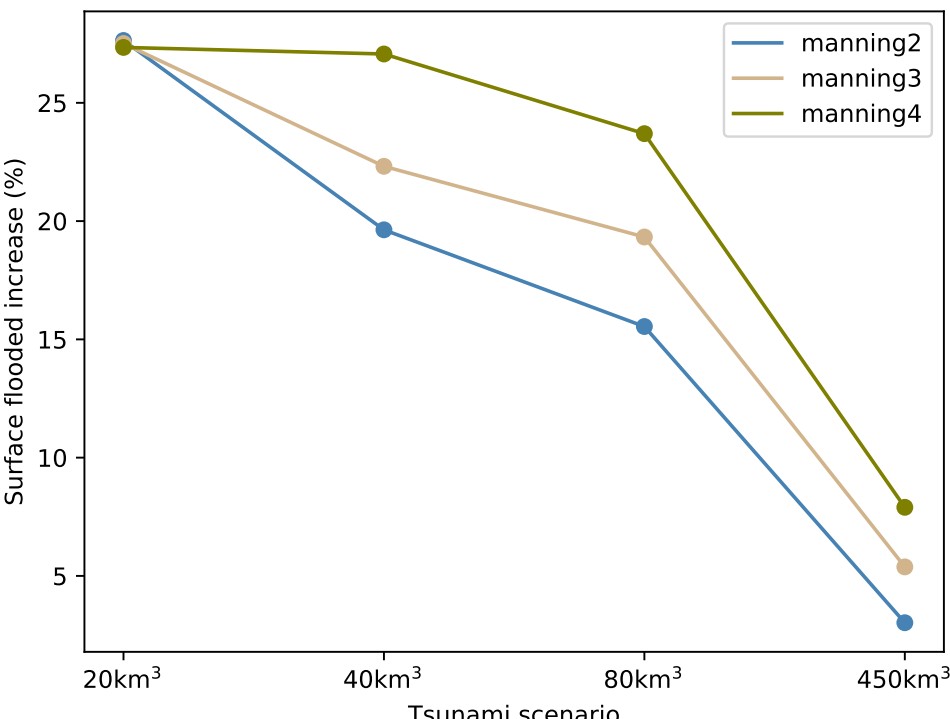

**Figure 12.** Percentage of change in flooded surface between the current situation and the sea level rise scenario and Manning configuration.

## 4. Discussion

The simulations of four potential tsunamis induced by remote landslides provide us with an overview of the most threatened places on the Guadeloupe archipelago. This series of simulations covers a wide range of impacts, from the lowest scenario in the current configuration to the most extreme scenario associated with climate change assumptions. A gradual impact is thus assessed in relation to gradual sources. These scenarios show very similar patterns of wave height distribution with a gradually increasing impact according to the volume of the slide. Wave heights follow a depth-induced distribution and are guided by refraction: wave trains are thus focused towards certain place that become very exposed. The two shelves off Saint-François and around Les Saintes (Figure 1) focus the energy towards Les Saintes, Saint-François, La Désirade and to a lesser extent toward Sainte-Anne (between Le Gosier and Saint-François). Le Moule is also exposed although no shelf is present off the coast on this side (east-north-east, see Figure 1).

Inundation level displays a linear increase between 20 and 80 km$^3$ and increases faster with a 450 km$^3$ to reach a maximum value of about 130 km$^2$. Although an event involving the simultaneous release of volumes up to 450 km$^3$ is regarded by many as unlikely [17,18], this extreme scenario cannot be totally ruled out and the simulations presented here show that such an event would have a catastrophic impact on the Guadeloupe, with waves of several tens of meters flooding most of the coastal areas. This type of scenario is largely above the worst case scenario anticipated so far in current safety plans [21].

The 20 km$^3$ scenario shows less striking water levels, but hazards may still be present in some places. This scenario needs to be anticipated with a high level of preparation and appropriate measures should be taken such as the evacuation of the immediate proximity of shore waters. This is particularly the case for the town of Terre-de-Haut, Les Saintes, which would be affected by the lowest simulated scenario. In this case, at least the first row of houses would be hit. This is quite a noteworthy result of these simulations as Les Saintes appear to be particularly exposed, although the town of Terre-de-Haut is facing west and therefore away from the direction of incoming waves. However, in all scenarios, refraction and convergence would increase water levels between the islands of Terre-de-

Haut and Terre-de-Bas and the town would be severely flooded in the event of a 40 km$^3$ slide. Elsewhere in the archipelago, the 40 km$^3$ case starts to trigger significant to severe floods in some places.

As the typical volumes of slide deposits observed in the vicinity of Canary volcanoes range from 50 to 200 km$^3$ [18], a volume of 80 km$^3$ could be taken as the reference case for the likeliest extreme flank collapse scenario. This potential volume would lead to a severe hazard scenario, threatening all coastal infrastructures. Important security measures are required and evacuation plans must be activated since water depth may reach above human height in several coastal towns such as Saint-François, Le Moule or La Désirade and Les Saintes. Such floods would undoubtedly induce large fluxes of water masses and hazardous currents.

In addition, as these scenarios cover a wide range of potential wave heights, they present a good opportunity to assess the consequences of climate change. The analysis of flooded surfaces gives a global assessment of the inundation and of a potential increase in future configurations from a minor to a major threat. Figure 12 shows that in all cases, the flooded surface will increase with sea level rise. However, it also highlights the fact that the impact of lower scenarios would increase more than that of heavier scenarios. This implies that sea level rise would lead to increased hazards, since lower scenarios are more likely to happen. The impact of the deterioration of the mangrove sees more effects with 40 and 80 km$^3$ slides. This is partially due to the fact that these scenarios are those that impact mangrove areas the most.

Globally, Grande-Terre is more subject to floods and to an increase of inundation under SLR scenarios than Basse-Terre, which is affected by fewer changes under future scenarios. This is mainly due to these island's different geological formation; Grande-Terre is made of slightly tilted flat deposits with vast low-lying coastal areas whereas Basse-Terre is a volcanic formation with steeper hill slopes (except for north-western part from Lamentin to Sainte Rose).

However, for all these simulations no specific validation could be performed for inundation modelling. The reliability of the inundation module only relies on earlier work presented by Zhang and Baptista [34]. Moreover, the definition of the friction coefficient is a crucial step that could significantly change the global flooded surface. Although particular attention was given to the definition of the friction coefficient according to land cover, the sensitivity of the inundated surface to the distribution of friction coefficient needs to be investigated further. Calibration against measurements would undoubtedly bring more reliability to the results. As a consequence, the flooded surfaces computed here are of interest as the basis for comparison between different scenarios and as a means of quantifying what consequences these differences could have, but they should not be taken as absolute values.

The use of and unstructured grid allowed us to achieve a high resolution along the shore-line and wherever land features needed to be represented with a high level of precision. However, due to computation time concerns, the resolution decreases quite fast as we move further inland where floods are less likely to occur. As a consequence, the level of accuracy decreases as the level of inundation increases and the simulation of propagation far inland is less accurate than in coastal areas.

Under the sea level rise scenarios, the adaptation of the mangrove or that of any other low lying back shore habitats has not been considered. We can imagine that during the next decades, as the sea level will rises continuously, the low-lying habitats will adapt their level and progressively follow the water elevation. Some areas could be filled and others may become eroded resulting in different forms of adaption according to the location. These complex changes have not been taken into account in this study, where we only introduced a rise of water level without any changes to the substratum level.

In addition, sediment transport is not taken into account and simple assumptions of runup, overtopping, overwash and rundown are modelled upon a nonmobile bed with

fixed bathymetry. This simple approach may limit the severity of the wave impact and the reach of inundation, as suggested by Tehranirad and Kirby [41].

The nesting between FUNWAVE and SCHISM was performed using a hotstart process, rather than by inputting water level and velocities time series at SCHISM boundaries. This implies that the last waves are travelling more with SCHISM than the first wave and therefore dispersive effects may be represented less accurately in the tail of the wave train.

Since inundation processes were simulated using a non linear but nondispersive model, interactions between nonlinearity and dispersion are not taken into account, and undular bores that may occur in some places are not considered. These undular bores and their breaking/propagation may affect the level of inundation [12,25]. However, taking them into consideration would require further model development which falls out of the scope of this study.

## 5. Conclusions

Using the most recent results of Abadie et al. [20] on the full or partial collapse of the Cumbre Vieja volcano and the associated wave it could generated, we simulated the propagation of the wave train towards the Guadeloupe archipelago to assess the tsunami's impact in the current configuration. Although the 20 km$^3$ scenario is not a major threat, its impact in some places may still be significant and appropriate measures of preparation should be planned even for this lowest scenario. From 40 km$^3$ the threat would become severe in some places and a volume of 80 km$^3$ can be considered as a severe hazard scenario with all coastal infrastructure potentially threatened.

Les Saintes are particularly exposed and vulnerable to the scenarios presented here even for the smallest collapse volume. The processes involved in the convergence and possible resonance need to be studied in more depth as the town of Terre-de-Haut requires particular attention.

These scenarios and simulations have also been used in this study to give an insight into the consequences of climate change, using assumptions on sea level rise and on the deterioration of natural protections. Results show that sea level rise would affect lower scenarios more than larger ones since the increase of the flooded area would be relatively more important for smaller surfaces. The experiment on the consequences of mangrove deterioration show that medium scenarios (40 and 80 km$^3$) are the most likely to be impacted but also reveal the model's high sensitivity to the use of friction coefficients which suffers from a lack of calibration. The flooded surfaces calculated here are therefore of interest for the purpose of comparison but should not be taken as absolute values.

The 450 km$^3$ collapse scenario is largely above the worst-case scenario anticipated in the current safety plans and although the probability of such an extreme scenario is regarded as small, the potential catastrophic consequences call for attention.

**Author Contributions:** Conceptualisation, G.E.A. and Y.K.; methodology, G.E.A.; software, G.E.A.; validation, G.E.A. and Y.K.; formal analysis, G.E.A.; investigation, G.E.A., Y.K. and N.Z.; resources, S.A. and B.D.; data curation, B.D.; writing—original draft preparation, G.E.A. and Y.K.; writing—review and editing, G.E.A. and S.A.; visualisation, G.E.A.; supervision, N.Z.; project administration, N.Z.; funding acquisition, N.Z. All authors have read and agreed to the published version of the manuscript.

**Funding:** This work, as a part of the project C3AF, was funded by the ERDF (Grant no: CR/16-115) and the Region Guadeloupe. The new tsunami source computations performed by S. Abadie have been performed in the framework of the PIA RSNR French program TANDEM (Grant no: ANR-11-RSNR-00023-01).

**Institutional Review Board Statement:** Excluded since the study did not involve humans or animals.

**Informed Consent Statement:** not applicable.

**Data Availability Statement:** The data presented in this study are available on request from the corresponding author.

**Acknowledgments:** Computational tests have been performed using Wahoo, the cluster of the Centre Commun de Calcul Intensif of the Université des Antilles. A large part of the graphical representation of the results were created with the open source library Matplotlib [42]. This work has been improved thanks to comments and suggestions of two anonymous reviewers. The authors are also very grateful to Valentine Leys who contributed in English editing and who made this text more clear to read.

**Conflicts of Interest:** The authors declare no conflict of interest.

## Abbreviations

The following abbreviations are used in this manuscript:

| | |
|---|---|
| C3AF | Consequences du Changement Climatique aux Antilles Françaises |
| SCHISM | Semi-implicit Cross-scale Hydroscience Integrated System Model |
| SLR | Sea level Rise |

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
