# Peer review of "How Would the Potential Collapse of the Cumbre Vieja Volcano in La Palma Canary Islands Impact the Guadeloupe Islands? Insights into the Consequences of Climate Change"

_geosciences, doi:10.3390/geosciences11020056_

Round 1

Reviewer 1 Report

Overall comment

This study estimated the tsunami hazard on the Guadeloupe Islands from the collapse of the Cumbre Vieja Volcano. The paper is well-documented in an appropriate manner using plain English. It was straightforward to follow their experiments, results, and discussion from the document. They applied The 3D Navier-Stokes, the 2D Boussinesq, and the Shallow water equations for generation, propagation, and inundation processes. These equations are appropriate to model the tsunami caused by the collapse of the volcano. The results may not be surprising, but they successfully demonstrated the tsunami hazard on the Guadeloupe Islands. The discussion is based on the numerical results and showed the limitations of the study. Their conclusion convinced us. I recommend that the editor accept the document for publication in Geoscience.

Minor comments
An "in" may be needed before Japan in Line2.
Characters for names of islands and stations in Figure 2 are hard to see. Please change the color.
In Line 137, please show references for the high-resolution LiDAR data.

Reviewer 2 Report

This is, in general a well  written document that provides a comprehensive estimate of the impact of a volcanically-generated tsunami in the Guadeloupe Archipelago in the Caribbean. The study addresses the far-field case of a volcanic explosion plus flank collapse of the Cumbre Vieja volcano in the Canary Islands.

The study also tries to incorporate the combined impact of Sea Level Rise (SLR) and the effects that this phenomenon may have on existing natural barriers (such as Mangroves) that may protect the Archipelago from tsunami impact of tsunami. These effects are accounted for by:

- correcting the present Sea Level to the estimated one by year 2100.

-by representing the presence or absence of natural barriers (Mangroves, Coral Reefs) with a modified, spatially varying friction coefficient.

The modeling approach is done in a reasonable way where the tsunami generation part of the event is not modeled but taken from previous work by Abadie et al., 2020) (4 potential scenarios of increasing magnitude are considered). This work included N-S modeling for the multiphase, slide-water interaction and dispersive effects in the generation regions were captured via Boussinesq modeling of the waves in the source region.

The trans-oceanic propagation is also modeled using a FUNWAVE-TVD, capable of capturing frequency dispersion as is recommended for this type of high frequency tsunamis in deep water.

While the inundation station was modeled using a version of SCHISM reduced to the NLSW equations.

In general, this is a rigorous and sensible approach where relevant physical processes in every stage are considered and the appropriate set of equations used to model the each stage. However there are still some considerations that I feel the authors should clarify for the reader:

1-Request: The inundation stage is performed using SCHISM, however with some assumptions in the RANS equations (and a single vertical layer) to reduce them to Non-Linear Shallow Water Wave (NSW) equations (l 123-125). While, I am of the opinion that dispersive effects generated in this last stage of tsunami modeling (inundation) are probably negligible, and the NSW equations are an appropriate mathematical model. Some authors think otherwise. For instance, the paper cites Lovholt et al., (lines 103-104) for the transoceanic propagation. A briefs explanation of why the inclusion of dispersive effects in the inundation stage is not necessary, would be appreciated.

2-Request: The authors refer to the process by which the solution of Abadie et al. 2020 is transferred to FUNWAVE, as a “hot-start” (l 117). It is unclear to me what is meant by this term. A better and more detailed description of it, would be appreciated by the reader.

3-Request: The results shown in Figures 4 and 5 show a remarkably linear behavior for the 4 source scenarios modeled on arrival at the inundation area. This is quite puzzling since one would expect dispersive effects in the generation region and trans-oceanic propagation to be more pronounced for smaller sources which tend to generate waves of higher frequency than in the case of larger sources, but no visible evidence of that effect is visible in the results.

Perhaps during transoceanic propagation, dominant wavelengths were long enough that dispersive effects were negligible, even for the smallest source?

Or if dispersive effects were indeed present, was perhaps the grid resolution used in the trans-oceanic propagation appropriate to capture the effects?

Some comment or clarification on this point would be helpful.

4-Request: The authors go to a large extent to model dry land terrain and submarine bottom characteristics with the use of a spatially variable friction coefficient, but they do not state whether the DEMs used in the models are bare-earth (as is standard practice) or include the built environment and vegetation canopy. If bare-earth, shouldn’t the built environment also be represented via an increased friction coefficient? Please, clarify this point.

The authors acknowledge the limitations and uncertainties inherent in their modeling as well as the lack of data for validation that could help reduce uncertainty in their estimates. This is something that I miss seeing in many papers and speak to the integrity and sense of reality of the authors.

-On a different level, while the paper is generally well-written with few grammatical errors. I think the writing style may not always be the most appropriate and results in some of the ideas not being conveyed efficiently. I would suggest an overall revision of the writing style to that of more standard English and general proof-reading of the manuscript.

Some examples:

-“This increase is more or less linear up to the scenario 80 km2  before increasing much more for the last scenario (450 km3 ) and represent respectively to the 20, 40, 80 and 450 km3  scenarios, 1.87, 9.99, 27.69 and 126.48 km 2  as shown on figure 5 for the current configuration of water level and mangrove coverage.” While I can distilled what it is trying to be said here, it is extremely fuzzy and confusing the way it is expressed.

-“The simulations of four potential tsunamis induced by remote landslides give an overview of the most threatened places on the Guadeloupe archipelago with a gradual impact from the smallest scenario in the actual configuration to the most extreme scenario within climate change assumption.”

Suggest:   Too many ideas mixed together. Perhaps break up into different sentences.

- “Elsewhere in the archipelago, the 40 km3  case, starts to see flooding to become significant with severe flooding in some places”

-Also, correct the words:

“corral” to “coral” (very different meaning). “

-…done thought...” to “…done through…”

-“The values of viscosity is…” to “…are…”

-“Their simulations are focused”   to   “…is focused”

-“to a lower extened... ”    to   “…to a lower extent…”
